# Atypical Sites of the Lipoma on the Hand and Fingers: Clinical and Imaging Features and Surgical Outcomes

**DOI:** 10.3390/diagnostics12102286

**Published:** 2022-09-22

**Authors:** Mihaela Pertea, Sorinel Lunca, Alexandru Filip, Dan Cristian Moraru, Claudiu Carp, Roxana Pinzaru, Vladimir Poroch, Bogdan Veliceasa

**Affiliations:** 1Faculty of Medicine, “Grigore T Popa” University of Medicine and Pharmacy, 700115 Iasi, Romania; 2Department of Plastic Surgery and Reconstructive Microsurgery, “Sf. Spiridon” Emergency County Hospital, 700111 Iasi, Romania; 3Second Surgery Clinic, Regional Institute of Oncology, 700483 Iasi, Romania; 4Department of Orthopaedics and Traumatology, “Sf. Spiridon” Emergency County Hospital, 700111 Iasi, Romania; 5Department of Palliative Care, Regional Institute of Oncology, 700483 Iasi, Romania

**Keywords:** lipoma, hand, finger, rare tumor

## Abstract

Background: Lipomas are the most frequent benign soft tissue tumor that are rarely found in the hand and are exceptionally rare on the fingers. The aim of this study was to investigate lipomas of atypical locations, so that they can be taken into account when making a differential diagnosis of a tumor of the hand or fingers. Methods: We studied a group of 27 patients diagnosed with lipoma of the wrist, palm, and fingers. The diagnosis was made by clinical and imaging examinations. Surgical treatment was established based on symptoms of pain, paresthesias, functional impairment, or aesthetic concerns. Treatment outcomes were assessed clinically at 1.5 years post-intervention. Results: Sizes over 5 cm were recorded in five cases, with the largest lipoma being 8 by 5 by 3.5 cm in size and weighing 125 g, located in the palm. There was one case of spontaneous tendon rupture and one case of carpal tunnel syndrome. There was no recurrence recorded at 1.5-year follow-up. Conclusions: Lipomas in the palm and fingers are rare entities (with more men affected), and surgical treatment consists of complete removal of the tumor and providing definitive healing. Despite their rarity, clinicians should consider lipomas when making differential diagnoses of soft tissue tumors of the hand.

## 1. Introduction

Lipomas are the most common form of soft tissue benign tumor in humans, accounting for approximately 16% of mesenchymal tumors [1]. These tumors can develop anywhere in the body, but most frequently occur in the upper body with 15–20% of cases located in the head and neck region [2]. The location of lipoma in the hand is rare, accounting for 1% to 3.8% of tumors in this anatomical region [3,4]. Lipomas of the fingers are extremely rare, with a reported incidence of 1% [5]. Lipomas may occur at any age but are more frequently encountered between the ages of 50–70, being rare in children [1,6,7]. Obese people are more prone to develop lipomas, and the incidence ratio of men to women seems to be equal [3,4,8,9]. Although composed of mature adipose tissue, lipomas are derived from mesenchymal preadipocytes [6]. Being slow-growing and often asymptomatic, lipomas of the wrist and palm exceeding 5 cm in size are reported.

Lipoma is defined as being “giant” when its dimensions exceed 10 cm and/or its weight exceeds 1000 g [9]. The malignant transformation of lipomas is uncertain, with giant ones being more prone to harboring malignant cells [6,8]. Recurrence is rare, being reported in approximately 5% of the cases because of incomplete surgical removal [10]. The aim of this article is to present the largest series of 27 cases of hand lipomas in the literature (15 cases of the palm, 9 cases of the fingers, and 3 cases of the wrist). Clinical features and diagnostic and surgical outcomes are analyzed and discussed. 

## 2. Materials and Methods

We studied a group of 27 patients, hospitalized and treated at the Plastic Surgery Clinic of County Emergency Hospital “Sf Spiridon” in Iasi, Romania between 2015–2020. The retrospective study was approved by the Ethics Committee of the “Sf. Spiridon” Emergency County Hospital. All patients were informed of the diagnosis and surgical treatment, with all patients giving informed consent. The diagnosis was made by clinical examination, using palpation to determine the consistency and mobility of the lesion and whether the pain was spontaneous or induced by palpation. Sensitivity was tested in all nerve areas. Active and passive motor activity of the wrist, hand, and finger segments were examined and compared with the contralateral hand. The Posch sign (hardening of the tumor after applying ice on it) appeared in all cases. Clinical examinations were supplemented by imaging investigations. Radiological and ultrasound examinations were performed in all cases. For large lipomas over 5 cm in size, MRIs were also completed to determine the exact relationships with neighboring structures, and to detect any malignant features. The decision for surgical treatment was established based on compression symptoms of a trunk or nerve branch (pain, paresthesias, functional impotence; finger flexion/extension impairment for mid-palm tumors; presence of carpal tunnel syndrome symptoms for lipomas at the wrist level), functional impotence caused by the large size of the tumor, or due to the unsightly appearance of the affected anatomical region, invoked by the patient. All 27 patients received surgery as curative treatment. Preoperatively, all patients underwent blood testing and blood cholesterol dosing. We did not perform a preoperative biopsy puncture in any of the cases. Surgery was performed under locoregional anesthesia (infraclavicular block), with a bloodless field achieved by applying the tourniquet to the arm in cases of wrist or palm lipomas and using the Wide-awake Local Anesthesia No Tourniquet (WALANT) technique with simple local anesthesia using Esmarch tape in those with finger lipomas. The surgery was performed by a senior plastic surgeon. 

In all cases the lipomas were well-encapsulated, the surgical technique consisting of excision by enucleation. For large lipomas, a drain was left in place for a few days. All resected tumors underwent a histological examination, which confirmed the diagnosis of lipoma. Follow-up was performed on a short-term basis at 2 weeks with suture removal, and long-term follow-ups with periodic evaluations were conducted at 3, 6, 12, and 18 months postoperatively by clinical examination.

## 3. Results

Of the patients in the study, 62.9% were male (17 patients) and ages ranged from 35 to 69 years (mean age 52 years). Lipomas were located in the wrist in three cases (11.1%), palm in fifteen cases (55.5%), and fingers in nine cases (33.3%). Of the fifteen cases with palmar location, there were two midpalm cases (13.3%) and thirteen cases (86.6%) in the thenar eminence. As for lipomas of the finger, one case (11.1%) was located on the volar aspect of the fingers and eight cases (88.8%) on the dorsal aspect. Only one case (11.1%) of lipoma was on the fourth finger. The index finger was involved in four cases (44.4%), and the middle and fifth finger in two cases each (22.2%) (Table 1 and Figure 1). 

The history of the disease ranged from 1 to 7 years. No patient had a history of trauma. Hypercholesterolemia was not reported in any of the patients. On palpation, a relatively mobile tumor mass with a soft consistency was detected in cases with wrist, thenar eminence, and finger location, whereas the tumor mass was fixed in cases with a midpalmar location. Hypoesthesia in the median nerve territory, without spontaneous or palpation-induced pain, was detected in three cases (11.1%) with wrist location. In the other cases, sensitivity was preserved and similar to that in the contralateral anatomical territory. Functional deficits were found in the cases of giant lipomas of the thenar eminence and those located in the midpalmar space, with a motor deficit in the execution of pinch movements, grasping, and mobility of the fingers. In one of the cases with a midpalmar space location, we detected the impossibility of fifth finger flexion and made the clinical diagnosis of spontaneous rupture of the superficial and deep flexor tendons (Figure 2). 

The Posch sign was positive in all cases. Improving the aesthetic appearance of the hand and fingers was invoked by the patients, in all cases. Radiological examination revealed the absence of skeletal changes and transparency of the soft-tissue tumor mass at different locations (Figure 3). 

In most cases, ultrasound examination confirmed the clinical diagnosis, with the lipoma appearing as a homogeneous, hyperechoic, and well-defined mass, without posterior enhancement or a Doppler signal. For giant lipomas (>5 cm), MRI scans were performed that showed a well-defined lesion with its own capsule, fine and homogeneous with a predominantly lipomatous signal (hypersignal on T1- and T2-weighted images and hyposignal on STIR images). 

Fine septa irregularly distributed inside the tumor with discrete gadolinophilia at the septa level were also described for these cases (Figure 4).

Complete lipoma excision was performed in all cases, these being well-encapsulated (Figure 5). 

The largest finger lipoma was the one located on the dorsal aspect of the fourth finger, measuring 3.5/1.7 cm (Figure 6). 

The largest sized lipoma in the study was recorded in midpalmar space, measuring 8/5/3.5 cm and weighing 125 g (Figure 7). 

The average sizes of the lipomas depending on location, measured by the excised pieces, were: 4.13/3.59 cm for tenar eminence, 6.25/4.5 cm for midpalm, 4.7/4.1 cm for wrist, and 1.85/1.21 cm for fingers.

The specimens removed during surgery were submitted for pathological examination, which confirmed the diagnosis of lipoma where no signs of malignancy were found, even for large tumors over 5 cm. In one case of a giant palmar lipoma, the patient underwent surgery again at 3 months for the reconstruction of the deep flexor of the 5th finger, damaged by lipomatous invasion. There were no intraoperative incidents or accidents and no immediate or remote complications in any of the cases. In all cases, removal of sutures was performed 14 days postoperatively. Postoperative occupational therapy was indicated for the patient with deep finger flexor reconstruction. The outcomes were good with full socio-professional reintegration of patients, who reported high satisfaction levels (Figure 8). 

Postoperative reassessments at one year and six months did not show any recurrence.

## 4. Discussion

Lipomas are the most common benign tumors in humans and third most common soft tissue tumor in the upper extremities [2,3,4,8,11,12]. However, hand and finger lipomas remain rare with few anecdotal reports or case series with a limited number of cases. Our case series of 27 hand lipomas represents the largest series to date in the literature. According to their location, lipomas can be dermal, subcutaneous, or subfascial. The 2002 World Health Organization classification of soft tissue tumors divided them into nine categories: lipoma, liposarcoma, lipomatosis of nerve, lipoblastoma, angiolipoma, myolipoma, chondroid lipoma, spindle cell pleomorphic lipoma, and hibernoma [12]. The etiology of lipomas is not well known. The genetic, metabolic, and traumatic causes have been discussed and lipomas were associated with the translocation and rearrangement of the 12q13-q15 and 6p13q chromosomal region [2]. Approximately 5% of patients diagnosed with lipoma have multiple lesions and a family history, the disease being inherited as an autosomal dominant trait [13,14]. No familial history or multiple site of lipoma were recorded for any of our patients. Diabetes was noted in two of our patients (7.4%), which may support a metabolic dysfunction etiology. The traumatic etiology of lipomas was explained by the stimulation of preadipocyte differentiation by fat necrosis and extravasation of blood due to trauma [15,16,17]. None of our patients had a positive history of trauma that could be related to the occurrence of lipoma. However, when we look at patient sex and the incidence of palm lipomas, we noticed that significantly more men were affected (11 men vs. 4 women). Although obvious severe traumas were not reported by the patients, these men are all manual workers, which we believe could support the theory that repetitive lower-grade trauma or microtrauma could be involved in the occurrence of lipomas at this level.

Although it is the most common benign tumor, incidence of lipomas in the hand and especially in the fingers is very rare. A possible explanation could be the presence of a smaller amount of adipose tissue in this region. The incidence of hand lipomas by sex is reported differently in the literature. Authors who report a higher incidence in women believe that this is because women have more adipose tissue [6]. However, significantly more men are affected by palm lipomas [6,8]. 

Our series comes to support a higher incidence in men, with 17 men vs. 10 women cases (62.9 vs. 37.1%). With regard to the anatomical segment, palm and wrist lipomas were more frequent in men (M:F = 13:5), and finger lipomas more common in women (M:F = 3:5). In a 2014 review, La Cruz et al. found36 cases of finger lipomas across 29 articles and were able to document 20 cases in women and 14 in men [18]. The long-term progression of lipoma is due to the slow and painless growth of this type of tumor. The compression exerted by lipoma on the neural elements and its large volume can cause sensitivity and pain disorders [19]. Leffert reported a case series of 141 lipomas of the upper extremities (from 122 patients) and recorded 109 asymptomatic and 32 symptomatic lipomas (26 painful and 6 with neurological symptoms) [4]. In one of our cases with a wrist lipoma, symptoms of carpal tunnel syndrome were present. Palmar lipomas generally have a thenar location [8,18,20]. Nadar (2010) reported thirteen cases of lipomas, of which nine were located in the hand, three in the fingers, and one in the forearm [8]. A study six cases of giant lipomas, located at the thenar eminence and palm, was conducted by Balvis in 2020 [21]. Zyluk (2021) reported a study on 40 patients with lipomas in the thoracic limb, with 11 of these being located at the wrist and hand level [3]. Kumar (2014) reported a series of three cases of hand lipomas and three in the fingers, with only fourteen cases recorded at that time in the literature [22]. In the current study, thirteen lipomas developed at the thenar eminence, two in the mediopalmar region, nine in the fingers, and three at the wrist level. 

Lipomas with a size greater than 10 cm are defined as “giant lipomas”, and are associated with a higher risk of malignancy [23,24,25]. A giant palmar lipoma infiltrates the deep layers of the hand [8,26,27]. Myhre-Jensen’s report of 1331 soft tissue tumors included 64 lipomas, which, in 95% of cases, were over 5 cm in size [28]. At the same time, 50% of the 72 soft tissue sarcomas reported in the same study were greater than 5 cm. Giant lipomas with a midpalm or thenar location can cause nerve elongation and sensitivity disorders, and also cause tendon rupture, as seen in one of our giant lipoma cases. It is worth mentioning that the size and weight of this palmar lipoma are unique to this study (8/5/3.5 cm and 125 g, respectively), as is the fact that it caused the rupturing of the fifth finger flexor tendons. 

Wrist lipomas can cause symptoms similar to carpal tunnel syndrome, with other etiologies. Clinical examination makes the diagnosis of lipoma in approximately 85% of cases. Radiological examination can confirm the diagnosis in approximately 70% of cases [19]. Most often, the combination of a clinical examination with radiological and ultrasound examinations leads to a diagnosis of lipoma. Regarding MRI scans, Capeslagui, in his study of 134 cases, showed that it had a predictability of 94%, and also determined the possible malignant nature of the lipomatous tumor [29]. Of the 27 patients reported in this paper, only five cases (18.5%) received a MRI scan, involving lipomas larger than 5 cm. The differential diagnosis of lipomas includes liposarcomas, which on an MRI appears as a heterogeneous mass, not clearly delimited by the adjacent tissues, invasive, and with indistinct margins. Cystic lymph nodes, leiomyomas, giant cell tumors, angiolipomas, glomus tumors, and sometimes neurofibromas can also be discussed in differential diagnosis [3,8,9,19,24]. Most often, tumors of the hand have a good prognosis. Malignant tumors of the hand are relatively rare, accounting for about 2% of all tumors [30]. Surgical excision is the mainstay of treatment for lipomas, regardless of their location and size. Surgical treatment of smaller lipomas is indicated to avoid complications due to tumor growth (nerve compressions, carpal tunnel syndrome, and sensitivity disorders). For giant lipomas, surgery is indicated for cosmetic reasons, symptoms (functional changes or sensitivity disorders), and risk of malignancy. Rydholm and Berg reported a series of 428 lipomas and determined a lipoma: liposarcoma ratio of 50:1 when tumor size was less than 5 cm and a ratio of 20:1 when lipoma size exceeded 5 cm [7]. When a lipoma is relatively small and located in the finger or other regions of the hand, it can be excised using local anesthesia in technique called WALANT [31,32]. Also beneficial are mesotherapy or intralesional injections of phosphatidylcholine or deoxycholate [33,34]. 

Lipomas can be also removed by liposuction or by endoscopic techniques [35,36]. None of these therapeutic options were used in the current study, as we opted for excision in all cases, with the lipomas being encapsulated and well-delimited. Recurrence of lipomas is rare, most often caused by incomplete excision. A recurrence rate of approximately 5% is reported in the literature [37,38]. Zyluk in his study on 40 upper limb lipomas, including 11 hand lipomas, were able to assess the surgical outcome during a mean of 4.2 years for 27 patients and found no recurrence [3]. In the current study, no lipoma recurrence was recorded at 1.5 years after surgery, implying that the tumors were completely removed in all cases. 

## 5. Conclusions

Lipomas of the hand are rare benign tumors. The current study reports the largest series of lipomas located at the level of the palm (fifteen cases) and fingers (nine cases). We noticed that significantly more men were affected with palm lipomas than women (11 men vs. 4 women), with all of these men, notably, being manual workers. A lipoma diagnosis can be established clinically, but for large, deep lipomas, an MRI is recommended to assist in preoperative planning. Neurological symptoms of carpal tunnel syndrome may be caused by the presence of a wrist lipoma. Large palm lipomas can cause nerve elongations and neurological symptoms, but also tendon erosions that can lead to spontaneous rupture of the flexor tendons of the fingers. Despite their rarity, clinicians should also consider lipomas in the differential diagnosis of soft tissue tumors of the hand. Often, surgical indication for asymptomatic hand lipomas will be given due to its unsightly appearance invoked by patients invoke. Surgery is the main treatment, with complete excision being possible in most patients and providing very good long-term results. In the case of a complete excision, recurrence is extremely rare. A long-term follow-up of more than 1–1.5 years is probably not necessary.

## Figures and Tables

**Figure 1 diagnostics-12-02286-f001:**
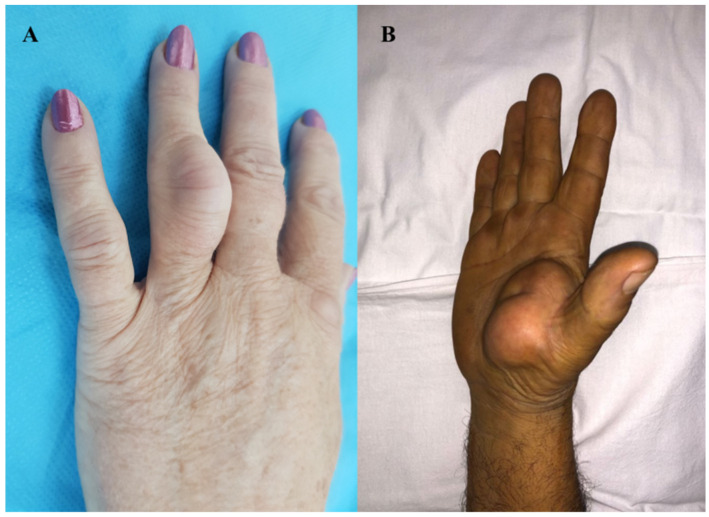
Clinical aspect of the lipoma: (**A**) ring finger localization, (**B**) thenar eminence localization.

**Figure 2 diagnostics-12-02286-f002:**
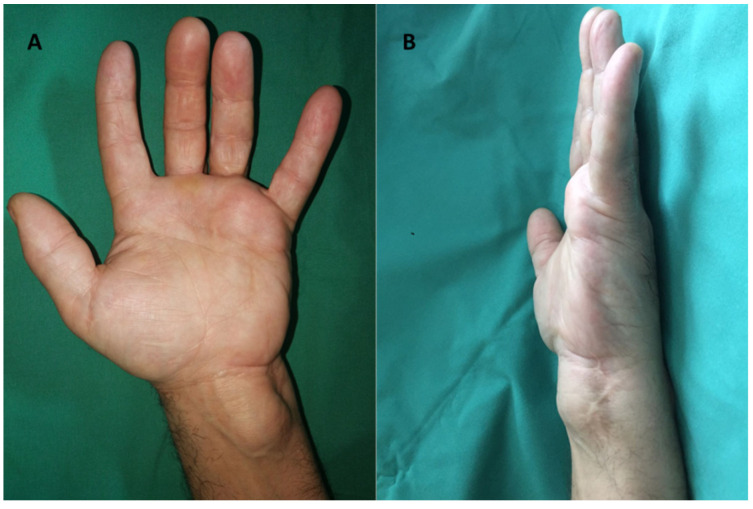
Giant lipoma in the midpalmar region and spontaneous rupture of the superficial and deep flexor tendons of the 5th finger: (**A**) palmar aspect and (**B**) lateral aspect.

**Figure 3 diagnostics-12-02286-f003:**
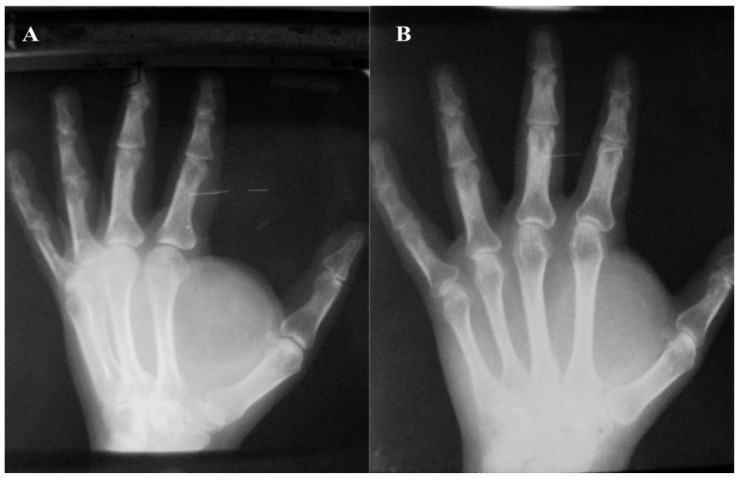
Lipoma in the hipothenar eminence—radiological aspect (**A**,**B**).

**Figure 4 diagnostics-12-02286-f004:**
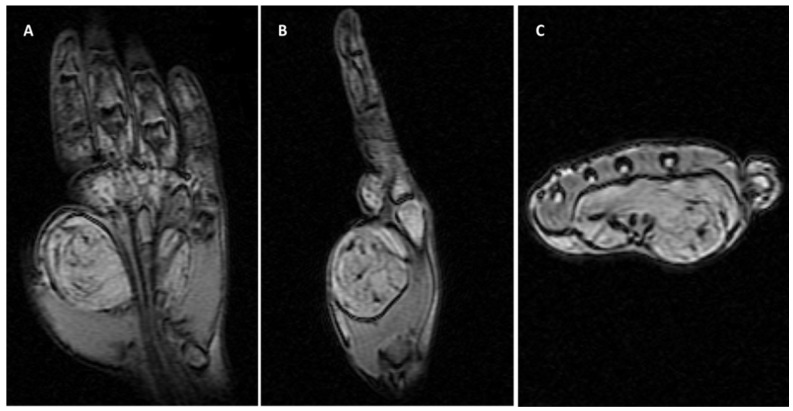
MRI of a giant midpalmar lipoma: clearly shaped lesion with fine homogeneous capsule, with predominantly lipomatosis signal and fine septa irregularly distributed inside the tumor with discrete gadolinophilia at septa level, beneath the palmar aponeurosis, with approximate size of 80/50 mm and a thickness of 35 mm. The lesion embedded and “detached” the group of tendinous flexors from the plan of the 2 interosseous muscles; the radial and ulnar artery with present flow in a superficial subaponeurotic location is in contact with the palmar surface of the lesion; the median nerve with a homogeneous hyposignal is structurally normal; the component bones of the wrist are normally configured, with homogeneous structure, regular bone contour, and cortically seamless; and the thenar and hypothenar muscle plans are compressed by injury (**A**–**C**).

**Figure 5 diagnostics-12-02286-f005:**
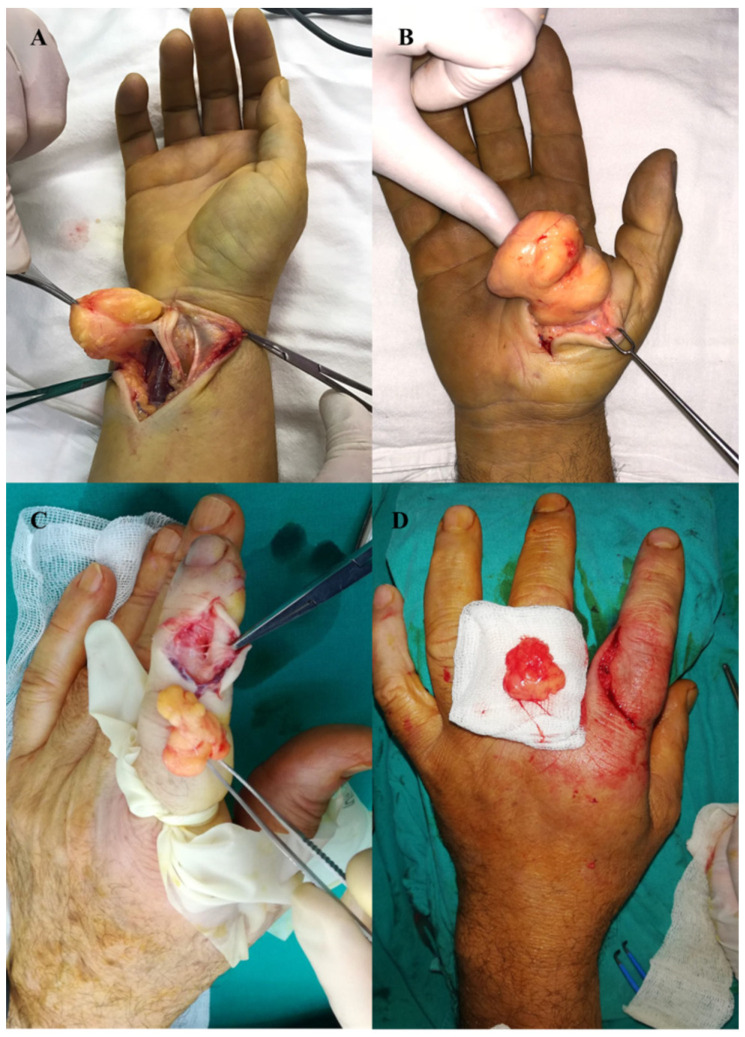
Intraoperative aspects of lipoma excision: (**A**) wrist, (**B**) thenar eminence, and (**C**) dorsal face of the index finger with removed specimen (**D**).

**Figure 6 diagnostics-12-02286-f006:**
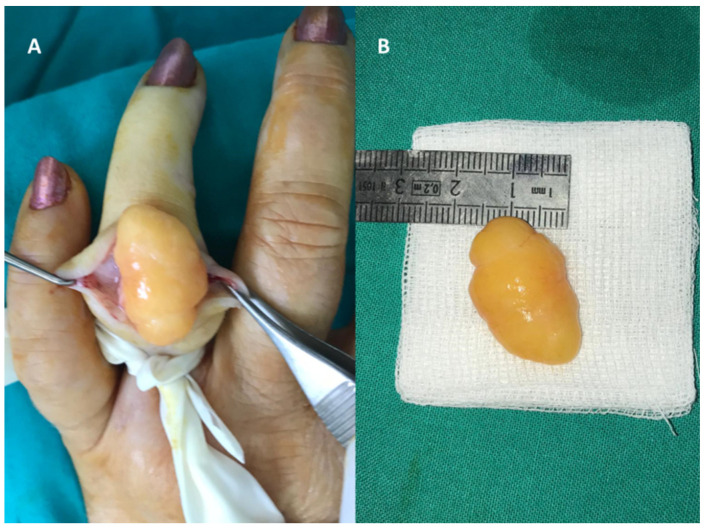
Large 4th finger lipoma measuring 3.7/1.7 cm: (**A**) intraoperative aspect and (**B**) removed specimen.

**Figure 7 diagnostics-12-02286-f007:**
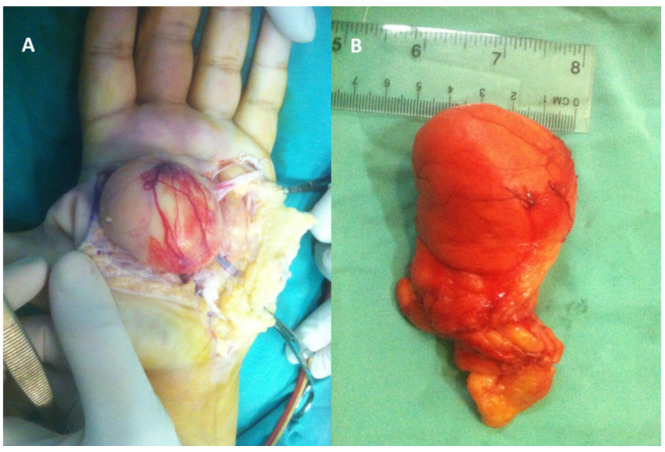
(**A**) Intraoperative aspects of a giant deep mid-palmar lipoma extended to thenar and hipothenar eminences—the thenar segment of lipoma was first approached and a tumor was located under the teguments, wrapped in a thin, well-vascularized fibrous capsule. During dissection, the tumor was found to be located under the palmar vascular arch and under common digital nerves, which were elongated; the incision was extended and the smaller portion of the tumor, located at the cubital hemihand, was approached, dissecting the tumor of tendinous adhesions. The 5th deep flexor tendon was found torn. Finally, the tumor was passed under the palmar vascular arch and extracted with the thenar segment through the first incision. (**B**) Removed specimen.

**Figure 8 diagnostics-12-02286-f008:**
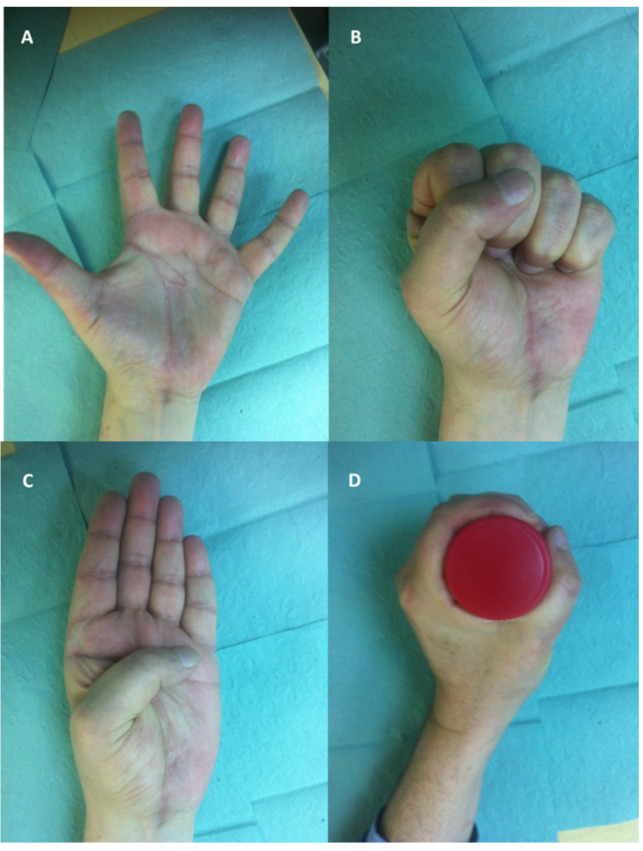
Functional and aesthetic outcomes after giant midpalmar lipoma excision and reconstruction of the 5th deep flexor: (**A**) flexion, (**B**) extension, (**C**) Kapandji score of 10, and (**D**) power grip.

**Table 1 diagnostics-12-02286-t001:** Characteristics of the study group.

	Age	Sex	Localization	History (y)	Size (cm)	Imaging Exam	Surgery	Pathology	Recurrence at 1.5 y
**1**	35	F	thenar eminence	3 y	5/4 cm	Rx/USG/MRI	complete excision	Lipoma	no
**2**	42	M	thenar eminence	2 y	4.3/4 cm	Rx/USG	complete excision	Lipoma	no
**3**	56	M	thenar eminence	2 ½ y	3/3.5 cm	Rx/USG	complete excision	Lipoma	no
**4**	43	F	thenar eminence	1 y	3/2.7 cm	Rx/USG	complete excision	Lipoma	no
**5**	49	M	thenar eminence	4 y	4.2/4 cm	Rx/USG	complete excision	Lipoma	no
**6**	58	M	thenar eminence	2 y	4.5/4 cm	Rx/USG	complete excision	Lipoma	no
**7**	52	F	thenar eminence	3 ½ y	4/4 cm	Rx/USG	complete excision	Lipoma	no
**8**	48	M	thenar eminence	3 y	4.5/4 cm	Rx/USG	complete excision	Lipoma	no
**9**	57	M	thenar eminence	5 y	5/4 cm	Rx/USG/MRI	complete excision	Lipoma	no
**10**	65	F	thenar eminence	3 y	5.2/5 cm	Rx/USG/MRI	complete excision	Lipoma	no
**11**	63	M	thenar eminence	2 y	4/3.5 cm	Rx/USG	complete excision	Lipoma	no
**12**	58	M	thenar eminence	3 ½ y	4/2 cm	Rx/USG	complete excision	Lipoma	no
**13**	53	M	thenar eminence	3 ½ y	3/2 cm	Rx/USG	complete excision	Lipoma	no
**14**	48	F	P1 dorsal face D5	1 ½ y	2/1.7 cm	Rx/USG	complete excision	Lipoma	no
**15**	69	F	P3 volar face D2	½ y	1/0.7 cm	Rx/USG	complete excision	Lipoma	no
**16**	57	M	P2 volar face D2	1 y	1.8/1.3 cm	Rx/USG	complete excision	Lipoma	no
**17**	67	F	P1 dorsal face D2	2 y	1.3/0.8 cm	Rx/USG	complete excision	Lipoma	no
**18**	44	M	P2 dorsal face D3	3 y	2.8/1.5 cm	Rx/USG	complete excision	Lipoma	no
**19**	39	F	P1 dorsal face D3	2 ½ y	1.5/1 cm	Rx/USG	complete excision	Lipoma	no
**20**	55	M	P1 dorsal face D2	2 y	1.8/1.5 cm	Rx/USG	complete excision	Lipoma	no
**21**	57	M	P1 dorsal face D5	1 ½ y	1/0.7 cm	Rx/USG	complete excision	Lipoma	no
**22**	47	F	P1 dorsal face D4	1 y	3.5/1.7 cm	Rx/USG	complete excision	Lipoma	no
**23**	48	M	mediopalmar	7 y	8/5 cm	Rx/USG/MRI	complete excision	Lipoma	no
**24**	39	M	mediopalmar	4 y	4.5/4 cm	Rx/USG	complete excision	Lipoma	no
**25**	44	M	carpal tunnel	2 y	5.2/4.8 cm	Rx/USG/MRI	complete excision	Lipoma	no
**26**	53	F	carpal tunnel	2 y	4.2/4 cm	Rx/USG	complete excision	Lipoma	no
**27**	59	M	carpal tunnel	1 y	4.7/3.5 cm	Rx/USG	complete excision	Lipoma	no

M = male, F = female, y = year, P = phalanx, D = digit, Rx = radiological exam, USG = ultrasonography, MRI = magnetic resonance imaging, cm = centimeters.

## Data Availability

Not applicable.

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
