# Peer review of "Atypical Sites of the Lipoma on the Hand and Fingers: Clinical and Imaging Features and Surgical Outcomes"

_diagnostics, 2022, doi:10.3390/diagnostics12102286_

Round 1

Reviewer 1 Report

An interesting manuscript on atypical sites of lipomas in the hand and fingers.

Introduction:

Please state the rate of transformation of lipomas to liposarcomas.

Materials:

Was it a prospective or retrospective study? Which type of imaging technologies did you use in lipomas < 5 cm to assess the location of the lipoma in respect to sensitive structures of the hand?

Please explain "Posch-sign".  Why did you use an Esmarche tape in WALANT procedures? If you wait long enough (min 30 Mins.) after injection there is hemostasis.

Were the surgical procedures performed by hand surgeons? Please state their Level of Experience (Tang et al.)

Results

Figure 3 is very reddish. Please change to grayscale.

Case in figure 7. "The 5th flexor tendon" -> "deep" is missing. Please state here if you reconstructed the tendon and how.

Line 189:  "ablation" -> removal of sutures.

Line 190:  "kynetotherapy" did you mean "occupational" therapy?

Please mention the sensitivity and more PROMs. A pictures of global hand funtion (fist clench) is not sufficient for a clinical report.

Discussion:

Line 224 to 226: consider revision. I suggest "However, significantly more men are affected (11 med vs. 4 women) by palm lipomas."

Line 231: delete "Why is this happening?"

Line 278: delete "Most".

Line 293: please state the recurrence rate depending on the technique used.

Author Response

Dear Reviewer,

Thank you for your appreciations and suggestions.

According to your recommendations, I made the following changes:

  • the rate of transformation of lipomas in liposarcomas is not very well difined
  • I mentioned that it is a retrospective study
  • I explained the Posch sign
  • I clarified that the Esmarch tape was not use in the Walant technique but in simple local anesthesia
  • I specified that the surgery were performed by a senior plastic surgeon
  • I modified Figure 3 according to the indications
  • I added “deep” to Figure 7
  • I wrote “removal sutures” and “occupational therapy”
  • to figure 8 I added the Kapandji scale and power grip
  • I added “However, significantly more men are affected by palm lipomas”
  • I deleted “Why is this happening? “and “most”

Thank you

Kind regards,

Mihaela  Pertea MD PhD

Reviewer 2 Report

The work is nice, but of low scientific innovation.

The only suggestion is to change the WHO of 2002 with a new classification of lipomatous tumors with the last edition of WHO

Author Response

Dear Reviewer,

Thank you for your appreciations and suggestions.

According to your recommendations, I made the following changes:

I changed the WHO of 2022 with a new classification of lipomatous tumors with the last edition of WHO.

Thank you,

Kind regard,

Mihaela Pertea MD PhD

Reviewer 3 Report

The authors study is  atypical locations of lipomas. The research design is appropriate and the results well explained and discussed.

I suggest to add in the abstract and in the conclusions the interisting difference that emerged between men and women: the incidence of sex for palm lipomas we notice that significantly more men are affected (11 men vs 4 women) with the importance of the manual work of these men.

Author Response

Dear Reviewer,

Thank you for your appreciations and suggestions.

According to your recommendations, I made the following changes:

I added in the abstract and in the conclusions :

“The incidence of the sex for palm lipomas we notice that significantly more men are affected with the importance of the manual work of these men”

Thank you,

Kind regards,

Mihaela Pertea  MD PhD